# Gene body methylation evolves during the sustained loss of parental care in the burying beetle

Peter Sarkies[1], Jennifer Westoby[2], Rebecca Mary Kilner [3] ✉ & Rahia Mashoodh [3,4] ✉

Epigenetic modifications, such as 5-methylcytosine (5mC), can sometimes be transmitted between generations, provoking speculation that epigenetic changes could play a role in adaptation and evolution. Here, we use experimental evolution to investigate how 5mC levels evolve in populations of biparental insect (*Nicrophorus vespilloides*) derived from a wild source population and maintained independently under different regimes of parental care in the lab. We show that 5mC levels in the transcribed regions of genes (gene bodies) diverge between populations that have been exposed to different levels of care for 30 generations. These changes in 5mC do not reflect changes in the levels of gene expression. However, the accumulation of 5mC within genes between populations is associated with reduced variability in gene expression within populations. Our results suggest that evolved change in 5mC could contribute to phenotypic evolution by influencing variability in gene expression in invertebrates.

There has been increasing interest in the idea that epigenetic mechanisms could participate in evolutionary change[1–5]. Epigenetic mechanisms are the biochemical marks (*e.g.*, DNA methylation, histone modifications, small non-coding RNAs) that can respond to environmental signals and regulate gene expression[6,7]. A hypothesised role for epigenetics in adaptation and evolution is driven primarily from recent work in plants and animals demonstrating the capacity for transgenerational epigenetic inheritance via epigenetic modifications[8–19]. This has led to speculation that epigenetic changes could act as drivers or initiators of the early stages of evolutionary adaptation[1,3] providing a possible mechanism for plasticity-led evolution[20,21]. Moreover, there have been several observations of population-level differences in epigenetic marks between closely-related species[9,13,18,22] and between locally adapted wild populations in a range of diverse taxa[10–17,19], which are often correlated with local abiotic environmental conditions.

The accumulation of epigenetic changes is suggestive of a potential role in adaptation to changing environments but lacking in plausible causal mechanistic evidence. There are a number of

pathways through which epigenetic marks could participate in adaptative processes and contribute to the emergence of population-level changes[1,5]. For example, epigenetic marks could be a by-product of transcriptional states or genetic sequence change[11,14,15]. In this scenario epigenetic modifications are effectors and/or markers rather than drivers of the adaptive change. Thus, epigenetic modifications could result from a complex interplay between processes acting in the short- and/or long-term that are environmentally-induced, genetic or stochastic (i.e., random variation generated when environments are the same; epimutational)[1,23]. Crucially, how and when such changes occur would depend on the persistence, heritability and function of specific epigenetic modifications and the genomic regions they target during adaptation, which vary across species[24–26].

DNA methylation, specifically 5-methylcytosine (5mC), is a highly conserved epigenetic mark across plants and animals. This involves the covalent addition of methyl groups to cytosine residues. In animals, 5mC is most prominent in the CG sequence context (CpG methylation; mCpG) which is added by DNA methyltransferases (DNMTs 1-3; e.g., for

[1]Department of Biochemistry, University of Oxford, Oxford, UK. [2]Department of Genetics, University of Cambridge, Cambridge, UK. [3]Department of Zoology, University of Cambridge, Cambridge, UK. [4]Centre for Biodiversity & Environment Research, Department of Genetics, Evolution and Environment, University College London, London, UK. ✉e-mail: rmk1002@cam.ac.uk; r.mashoodh@ucl.ac.uk

maintenance during cell division and de novo) and, in mammals, can be actively removed by ten-eleven translocation enzymes (TETs)[27]. Across arthropods, genome-wide levels of methylation tend to be generally lower than in vertebrates and are restricted to a subset of genes[24,28,29]. 5mC is predominantly found in the transcribed regions of genes, known as gene bodies suggesting important, yet still poorly understood, roles for 5mC in the species that carry this epigenetic modification[28,30]. Genes that possess high levels of 5mC are more likely to be moderately expressed housekeeping genes and exhibit well-positioned nucleosomes around the promoter region[29,31]. The high heritability of insect gene body methylation has led to speculation that it might be particularly sensitive to evolutionary change[32] as has been shown in plants and other animals[11,19,23,26]. However, few studies have tested this explicitly. Moreover, many studies have failed to find a causal link between levels of gene expression and changes in 5mC that readily occur in response to various environmental conditions[29,33–36], leading to the conclusion that environmentally-induced changes in gene body 5mC have limited effects on transcriptional outcomes in insects. Instead, it has been suggested that DNMTs in insects perform important functions in germline development, independent of 5mC control of transcription[37–41].

Here we use experimental evolution to examine how patterns of gene expression and DNA methylation (5mC) are induced and evolve in populations of burying beetles as they adapt to a change in their social environment, namely the removal post-hatching parental care. Burying beetles possess single copies of DNMT1 and 3, which comprise a functioning 5mC system with levels of 5mC that are comparable to other arthropod species and[29,42,43]. Parental care (and its loss) has been associated with plasticity in both 5mC across multiple different genomic features including CpG islands, promoter/enhancer regions and transposable elements (TEs) and gene expression across a number of different species[33,44–47]. However, its role in insect 5mC and gene expression plasticity is still poorly understood.

In natural populations, burying beetle parents raise their young on a carrion nest formed from a small dead animal, such as a mouse or songbird. There is continuous variation in the level of post-hatching parental care supplied[48]. At one extreme, parents tend to their offspring throughout their development, whereas at the other extreme, parents abandon their offspring before hatching[49]. We exploited this natural variation in care to establish two types of experimentally evolving populations in the laboratory (each replicated twice), which varied only in the family environment that larvae experience during development, and where the same family environment was created for successive generations within populations. In Full Care populations (FC_POP), parents remained with their young throughout development; whereas in No Care populations (NC_POP), parents were removed just before their offspring hatched[50]. Previous work showed that that NC_POP populations rapidly adapted to a life without parental care such that by generation 13 the proportion of successful broods matched that of the FC_POP populations[50]. Adaptation to the loss of parental care was associated with a number of changes in morphology and behaviour. For example, NC offspring evolved relatively larger mandibles perhaps to help with feeding on the carcass in the absence of parents[48], they hatched more synchronously[48] and they cooperated more with each other[51]. Moreover, our whole-genome sequencing analysis of these populations indicates that that loss of care induced strong directional selection and divergence at multiple gene loci[52].

Here, we used experimental evolution to analyse the causes and functional consequences of evolved changes in gene body 5mC and investigate its potential role in adaptive evolution and behavioural change. We characterised the extent to which epigenetic changes respond to an initial change in the environment, and investigated the degree to which such changes might persist and be correlated with changes in gene expression in populations adapted to a changed social environment. To distinguish between evolved and environmentally-induced changes, we used a common garden approach. Offspring from the evolving populations were exposed to the reciprocal parental environment, as well as to the parental environment under which they had evolved (see Fig. 1a for experimental design summary). This enabled us to determine whether gene body 5mC levels evolve in response to the experimental loss of post-hatching care, and if so, whether these changes were associated with population differences in transcription (either absolute levels of RNA or variability in RNA expression between individuals) and/or genetic sequence divergence at methylated genes. Here, we show that changes in 5mC that accrue between populations is associated with reduced variability in gene expression within populations, rather than changes in the level of gene expression between populations.

## Results
### Differential gene expression in response to the sustained removal of care
Differences in gene expression between the populations could be due to evolved differences (stable) or the recurrent exposure of the different care regimes imposed on the populations, which could be transient and/or reversible. To distinguish between these alternatives, we sampled populations in their native (FC_POP FC_ENV and NC_POP NC_ENV) as well as reciprocal environments (FC_POP NC_ENV and NC_POP NC_ENV) (see Methods). Since all beetles in this study were derived from the same source population[50,52], this design also enabled us to use the response of the FC_POP in a NC_ENV as a proxy for the initial response to the loss of care, allowing us to compare it to the response to NC_ENV after 29 generations of experimental evolution (NC_POP). To characterise the source of genome-wide variation in first-instar larval gene expression between our populations quantitatively, we conducted a discriminant analysis of principal components (DAPC), which is an extension of principal component (PC) analyses that collapses PCs into a single measure of variation, using all expressed genes ($n = 12{,}772$) within all samples. The removal of care was significantly associated with a shift in gene expression (environment posterior mean = 4.22, 95% CI = [3.35, 5.15], $p_{MCMC} < 0.001$; Fig. 1b) and this response was true whether individuals were derived from the FC or NC populations, as there was no effect of population (population posterior mean = 0.27, 95% CI = [−0.63, 1.20], $p_{MCMC} = 0.576$) nor an interaction between population of origin and the current environment (posterior mean = −7.91, 95% CI = [−2.12, 0.40], $p_{MCMC} = 0.210$). Further, each replicate responded to the care treatment in a similar way, since there was no effect of block on gene expression plasticity (posterior mean = 0.03, 95% CI = [−0.6, 0.72], $p_{MCMC} = 0.940$). Taken together, these results suggest that while there were large environmental shifts in gene expression in response to the initial loss of parental care, evolved changes in gene expression in response to the sustained loss of care were likely to be more subtle and/or involve smaller subsets of genes. For example, adaptation to the loss of care following experimental evolution could involve a change in the total number of genes being responsive to the loss of care and/or the same genes changing in the magnitude of their response to the loss of care.

To investigate these possible differences in more detail, we performed differential expression analyses using FC_POP FC_ENV as the reference population to estimate the number of differentially expressed genes (DEGs) across conditions. We found that 657 genes were differentially expressed when parental care was removed for the first time in the FC_POP ($n = 657$; all $\log_2$ fold changes > 1; Fig. 1c). By contrast, in the NC_POP, which had experienced a NC_ENV for the previous 29 generations, 385 genes were differentially expressed which is a significant reduction in response ($X^2(3, n = 12{,}772) = 21{,}558$, $p < 2 \times 10^{-16}$). Therefore, the response to the loss of care involved fewer DEGs in the NC_POP than FC_POP suggesting the response to the loss of care was blunted following experimental evolution. Interestingly, when NC_POP parents were allowed to care for their offspring, there were minor

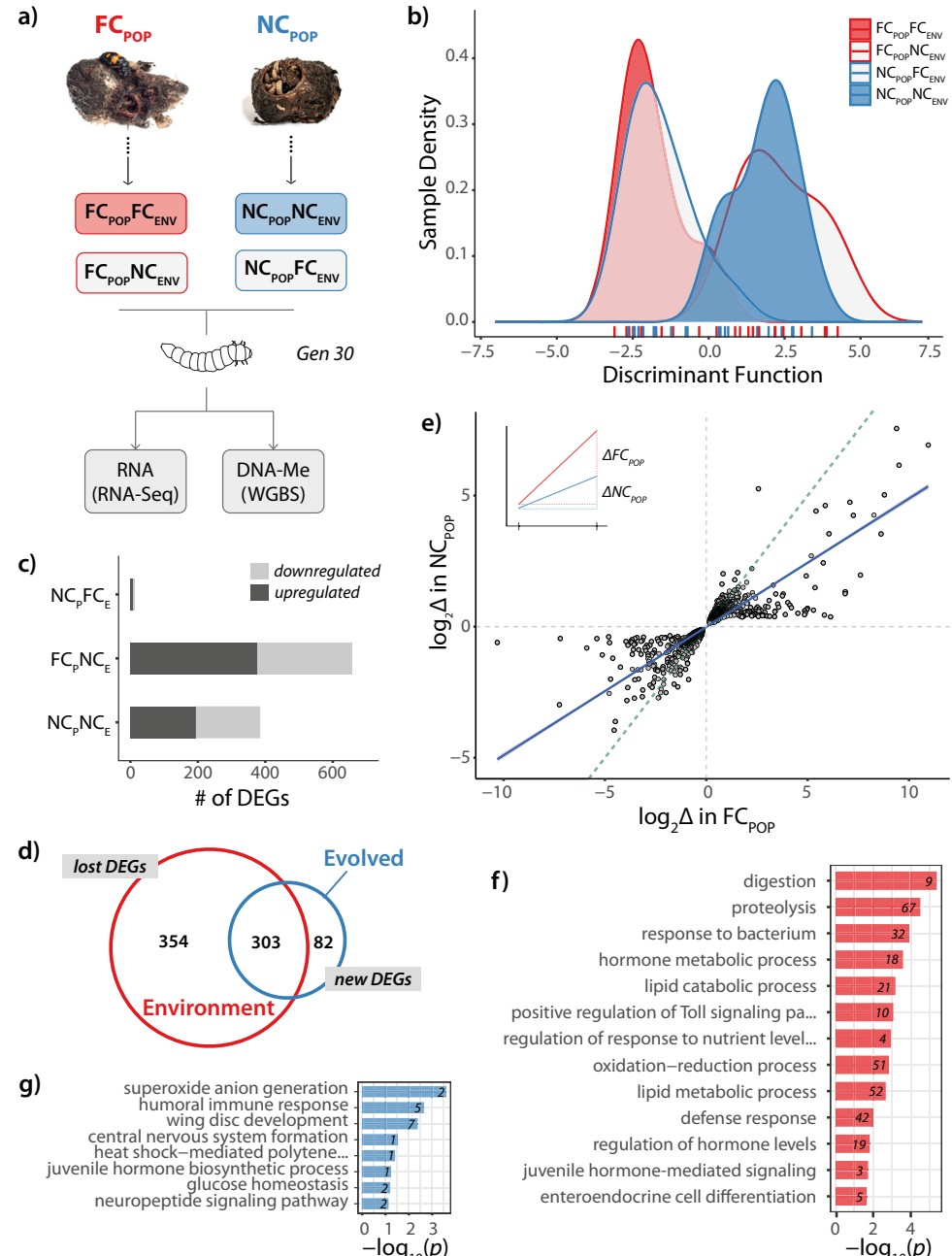

**Fig. 1 | Gene expression diverges in response to the sustained removal of care.**
**a** Summary of sampling from experimentally evolving populations and sequencing strategy. Larval heads from Full Care ($FC_{POP}$) and No Care ($NC_{POP}$) populations in native ($FC_{POP}FC_{ENV}$; $NC_{POP}NC_{ENV}$) and reciprocal environments ($FC_{POP}NC_{ENV}$; $NC_{POP}FC_{ENV}$) were sampled and used to measure gene expression (RNA-Seq) and DNA methylation (whole-genome bisulfite sequencing; WGBS). **b** Density plots of discriminant function values for each family indicating that variation in gene expression is primarily due to the current social environment ($p < 0.0001$; generalised linear model; $n = 12$ biological replicates per condition). **c** Number of differentially expressed genes (DEGs) relative to the $FC_{POP}FC_{ENV}$ group. **d** Venn

diagram indicating degree of overlap between DEGs induced after one generation's exposure to $NC_{ENV}$ (environmental) versus 30 generations exposure to $NC_{ENV}$ (evolved). **e** Scatterplot showing the correlation between log-fold changes ($\log_2\Delta$) in the response to a $NC_{ENV}$ within each population for overlapping DEGs (slope = 0.489, $p < 2.2 \times 10^{-16}$; generalised linear model). **f** Gene ontology (GO; biological processes) enrichment for a single generation upon exposure to a $NC_{ENV}$ and **g** GO terms for "new" DEGs after 30 generation's exposure to $NC_{ENV}$. Only GO terms with FDR-corrected $p < 0.05$ are shown (Fisher's exact tests). Numbers indicate a number of genes associated with that category.

differences (only 20 DEGs) when compared to the $FC_{POP}FC_{ENV}$, suggesting that gene expression in the $NC_{POP}$ was not constrained by their exposure to divergent selection pressures during these generations.

Moreover, there was a significant overlap in identity of the genes being differentially expressed in $FC_{POP}NC_{ENV}$ compared to the $NC_{POP}NC_{ENV}$ ($\log_2(OR) = 122.4265$, $p < 2.2 \times 10^{-16}$), with many more changes being lost ($n = 354$) than being gained ($n = 82$; Fig. 1d;

Supplementary Data 1). To test whether this reflected possible changes in the overall strength of response by individual genes to a NC environment within each population, we extracted overlapping DEGs common to $FC_{POP}NC_{ENV}$ and $NC_{POP}NC_{ENV}$ (regardless of their log-fold change) and correlated the magnitude of environmental response (i.e., log-fold change when care is lost) within each population for each individual gene. This is the equivalent of comparing reaction norms on

a genome-wide scale (See Fig. 1e inset). We predicted that if the magnitude of change in response to the loss of care was similar across populations, then we would expect fold-changes across overlapping genes to be highly correlated between the two populations, resulting in a slope close to 1. We found that genes that were differentially expressed in response to a $NC_{ENV}$ in both populations were positively correlated, with no evidence that genes switched direction of expression after evolving without parental care (slope = 0.489, $t(3103) = 91.875$, $p < 2.2 \times 10^{-16}$; Fig. 1e). However, the slope of this correlation was significantly less than a slope of 1 ($F(2,3103) = 9176.4$, $p < 2.2 \times 10^{-16}$). Taken together, this suggests that while there were similar gene sets affected by the removal of care across both populations, the magnitude of response was consistently lower in the $NC_{POP}$ compared to the $FC_{POP}$ at those genes.

We extracted significant gene ontology (GO) terms from each contrast and collapsed across significant, but highly redundant terms, to look for overarching patterns. This analysis revealed that changes in gene expression primarily occurred in three broad categories of genes: 1) stress and its cellular response, 2) immune function and 3) growth and development (see Fig. 1f for examples of GOs falling within these categories). These results suggest that whilst the initial response to the removal of care is associated with an upregulated stress response and increased immune defence, individuals selected under a NC regime expressed fewer genes associated with these two categories. Moreover, in the $NC_{POP}$, these changes were accompanied by enhanced expression of genes associated with physiological and neurobiological development (Fig. 1g). Further examination of gene expression changes that emerged after 30 generations of exposure to a NC environment revealed an over-representation in the neurotransmission and neuropeptide categories (*e.g.*, neuropeptide Y, orexin, and glutamate). For a full list of GO categories, see Supplementary Data 2.

## Differential methylation within genes in response to the sustained removal of care

We analysed methylomes of first instar burying beetle larvae and confirmed that 5mC was predominantly enriched in gene bodies and associated with genes with moderate to high transcription (see Supplementary Note and Figs. S1–S2). We, therefore, chose to focus on 5mC in gene bodies for subsequent differential methylation analyses. We performed analysis of differential 5mC levels across gene bodies (weighted mean of 5mC sites across a gene) to compare[1] levels of 5mC between populations (evolved changes; sustained exposure to a $NC_{ENV}$ over 30 generations) with[2] 5mC levels induced by a single generation of exposure to a $NC_{ENV}$ within the $FC_{POP}$ (proxy for initial response) and[3] 5mC within the $NC_{POP}$ when exposed to a $FC_{ENV}$ (population-specific responses to $NC_{ENV}$). As has been previously reported, not all genes in the burying beetle have 5mC[29,43]. Therefore, we clustered average weighted 5mC (calculated across the entire gene body), identified two distinct distributions of 5mC states within genes (see Methods) and focussed our analyses on those that were classified as 5mC-containing (Fig. S3a, $n = 4431$; 39% of genes for which 5mC was present for 85% of samples). These genes were also more highly expressed ($W = 9161933$, $p < 2.2 \times 10^{-16}$, Fig. S3b). Consistent with previous findings[29], genes orthologous to housekeeping genes in *Drosophila melanogaster* displayed higher 5mC levels than those that were not housekeeping gene orthologs ($p < 2.2 \times 10^{-16}$, Wilcoxon Test; Fig. S3).

We first compared the environmental response to a NC environment (within the $FC_{POP}$) to the response 30 generations later (evolved differences). Removal of care in response to one generation of a $NC_{ENV}$ resulted in 1486 genes with altered 5mC levels, whereas we found 1530 differences when comparing the evolved response to the sustained loss of care (all genes FDR-corrected $p < 0.01$). We also assessed the overall false discovery rate empirically by randomly shuffling samples between conditions 1000 times and extracting the number of

differentially methylated genes (DMGs) after FDR correction at each shuffle. The number of DMGs detected in each comparison far exceeded the number that would be expected at random ($p_{FDR} = 0$ for both comparisons; Fig. 2a). Moreover, although there was some overlap in response to the removal of care within each population (overlap of 741 DMGs; $\log_2(OR) = 1.8$, $p < 2.2 \times 10^{-16}$), a proportion of these changes remained specific to populations (Fig. S5). Taken together, this suggests that 5mC levels likely arise through both environmental and population-specific differences.

## Differential levels of 5mC within genes and transcriptional change

Of the DMGs that were altered in response to a single generation of exposure to NC, 51% were associated with differences in gene expression at those genes ($\log_2$(odds-ratio; OR) = 1.21, $p = 0.003$). However, after 30 generations, 49% of DMGs were associated with gene expression differences and this was not significant ($\log_2(OR) = 1.16$, p = 0.27). Both overlaps occurred at chance levels (49% vs 51%), suggesting limited associations between differential levels of 5mC (DMGs) and gene expression (DEGs), and any differences in p-value reflected the extreme response in gene expression (i.e., more DEGs) when care was lost for the first time in the $FC_{POP}$. GO term enrichment analysis revealed that genes with different 5mC levels after a single exposure to the loss of care were enriched in stress and growth reduction pathways (Fig. 2d). Genes with different levels of 5mC in the evolved response to a $NC_{ENV}$ were enriched in biological processes related to growth and brain development (Fig. 2e; see Supplementary Data 2 for a full list of GO terms). Furthermore, genes with differential 5mC levels and differentially expressed in response to the removal of care (environmental shift) were significantly enriched for housekeeping genes compared to evolved changes (FDR-corrected $p < 0.05$, Fisher's Test; Fig. 2f). Interestingly, this enrichment was lost in genes that were differentially expressed when comparing the two evolved populations (Fig. 2f). Given the high degree of overlap in DMGs induced by the environment compared to the evolved response to the $NC_{ENV}$ (48%; Fig. 2b), we hypothesised that the latter set of changes might reflect environmentally-induced changes that persist despite changes in expression becoming dampened (i.e., 5mC levels reflect ancestral rather than current transcriptional state).

To investigate this idea further, we highlighted genes with different 5mC levels on density plots describing the relationship between expression (FKPM) and average methylation for all genes. We identified a cluster of genes that showed different levels of 5mC but that had lost differential expression after 30 generations of sustained exposure to NC (Fig. 2c). Differential 5mC in this cluster (outlined in Fig. 2c) is associated with differential gene expression in populations exposed to NC for one generation ($\log_2(OR) = 1.91$, $p = 0.01$) but showed no association in the evolved response ($\log_2(OR) = 0$, $p = 4.279e-05$). These genes also had different 5mC levels at this cluster when the $NC_{POP}$ were exposed back to the $FC_{ENV}$ for a single generation suggesting they are environmentally induced (i.e., not inherited genetically) (Fig. S6). We performed a GO analysis on this cluster and found these genes were almost exclusively enriched in ribosome biogenesis, protein translation and transport processes (see Fig. 2f and Supplementary Data 2). This further suggests that some DMGs can be transient, while other changes can persist, which do not appear to correspond to changes in gene expression induced by the persistent removal of care. In other words, changes in 5mC occur independent of changes in transcription levels.

We then asked if DMGs that were initially responsive to the environment could become fixed evolved differences, that were no longer environmentally sensitive. We reasoned that such changes would not be reversed when $NC_{POP}$ were provided with a $FC_{ENV}$. There were 273 DMGs (18% of all evolved DMGs) that were differentially methylated in the evolved populations that were not reverted by

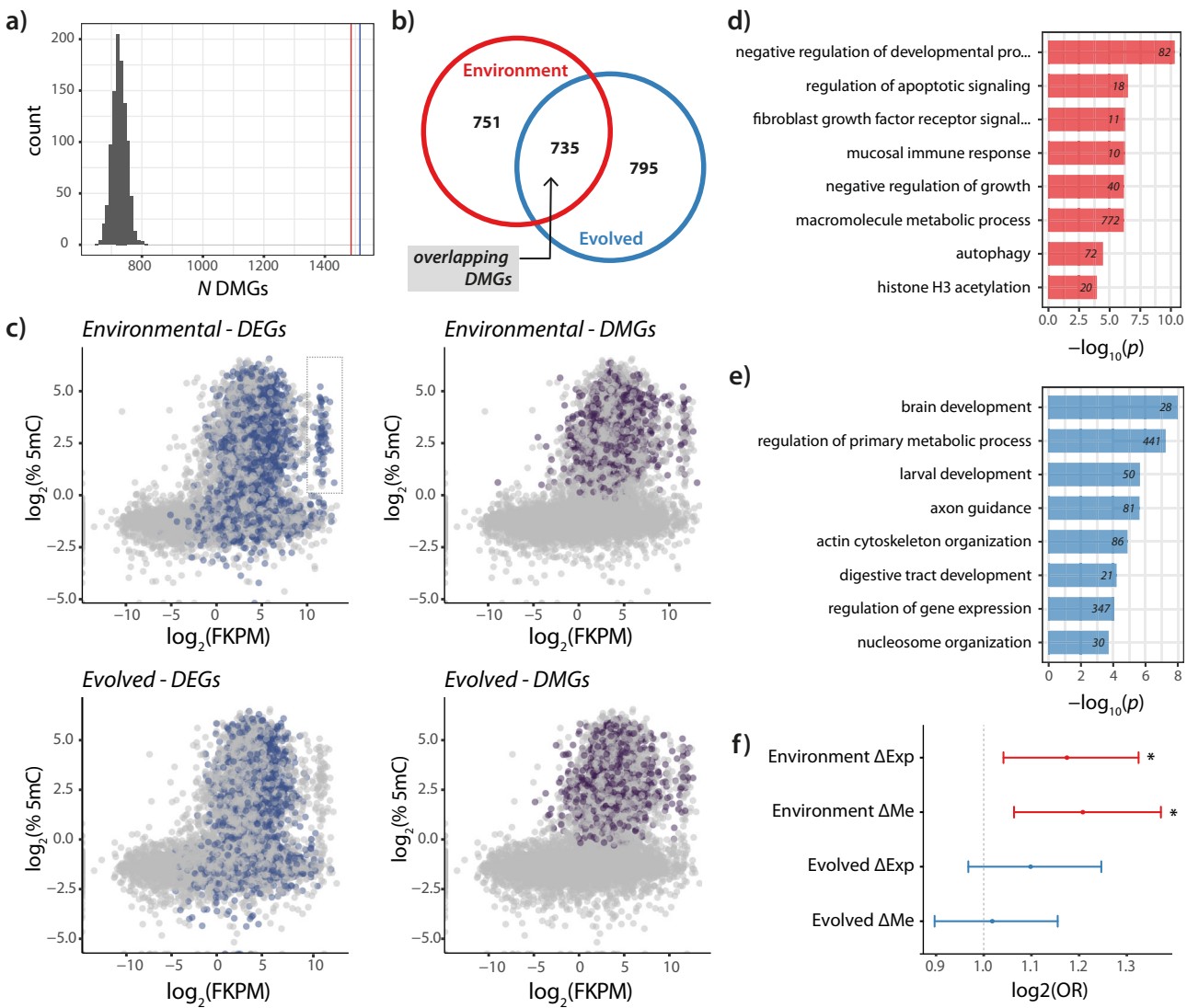

**Fig. 2 | 5mC levels of gene bodies diverges in response to the sustained removal of care. a** Histogram of number of differentially methylated genes (DMGs) passing FDR-correction when samples are shuffled randomly (binomial generalised linear model). Vertical lines indicate the number of DMGs found between environmental (single exposure to $NC_{ENV}$; red; n = 7 biological replicates) and evolved contrasts (30 generations of exposure to $NC_{ENV}$; blue; n = 7 biological replicates). **b** Venn diagram indicating the degree of overlap between environmental and evolved DMGs. **c** The relationship between gene expression (FKPM) and percent CpG methylation (% 5mC) for all genes analysed for environmental (top) and evolved (bottom) contrasts. Colours indicated differentially expressed genes (DEGs; blue) and DMGs (purple) for each contrast. **d** Gene ontology (GO; biological processes) enrichment for response to the loss of care DMGs after one generation's exposure to $NC_{ENV}$ (Fisher's exact tests) and (**e**) after 30 generations of exposure to $NC_{ENV}$. Only GO terms with FDR-corrected $p < 0.05$ are shown (Fisher's exact tests). Numbers indicate number of genes associated with that category. **f** Enrichment scores (and 95% confidence intervals) for associations between differential expression (ΔExp) and differential methylation (ΔMe) in housekeeping genes (Fishers exact test, *indicates FDR-corrected $p < 0.0001$).

exposure of the $NC_{POP}$ to a $FC_{ENV}$. While these genes were enriched in defense, cellular and RNA metabolic processes, they were not significantly associated with gene expression changes in larvae ($\log_2(OR) = 0.97$, $p$ = ns). This raises the possibility that some heritable changes in 5mC levels could become fixed during adaptive divergence, whereas others are due to recurrent induction by the same environment for generation after generation. Regardless of their origin, changes in 5mC levels did not appear to be associated with evolved changes in gene expression levels.

## Increased 5mC at genes is associated with reduced variability in gene expression

The preceding analyses showed that methylation levels diverged between populations exposed to different regimes of care. However, such changes in 5mC could reflect changes at a few random individual

CpGs or a co-ordinated change across multiple CpGs within the same gene. The latter could reflect a gene-level process governing shifts in gene methylation which could have implications for function and the scope for adaptation. This raises the possibility that, in addition to 5mC levels varying across populations within a core subset of genes, genes might also acquire and/or lose methylation over evolutionary time. To identify such genes directly, we first identified which genes possessed 5mC in the two populations separately (see Methods; Fig. S7). There was a high degree of overlap in the genes that were consistently classified as having 5mC in both populations ($NC_{POP}$: $n = 4215$, $FC_{POP}$: $n = 4189$); however, we identified a small fraction of genes that had 5mC in $FC_{POP}$ and not in NC ($n = 171$), and a smaller number that had 5mC in $NC_{POP}$ and not in $FC_{POP}$ ($n = 217$). We then sought to determine whether such changes were specific to evolved versus environmental conditions. We reasoned that if 5mC levels changed in a coordinated

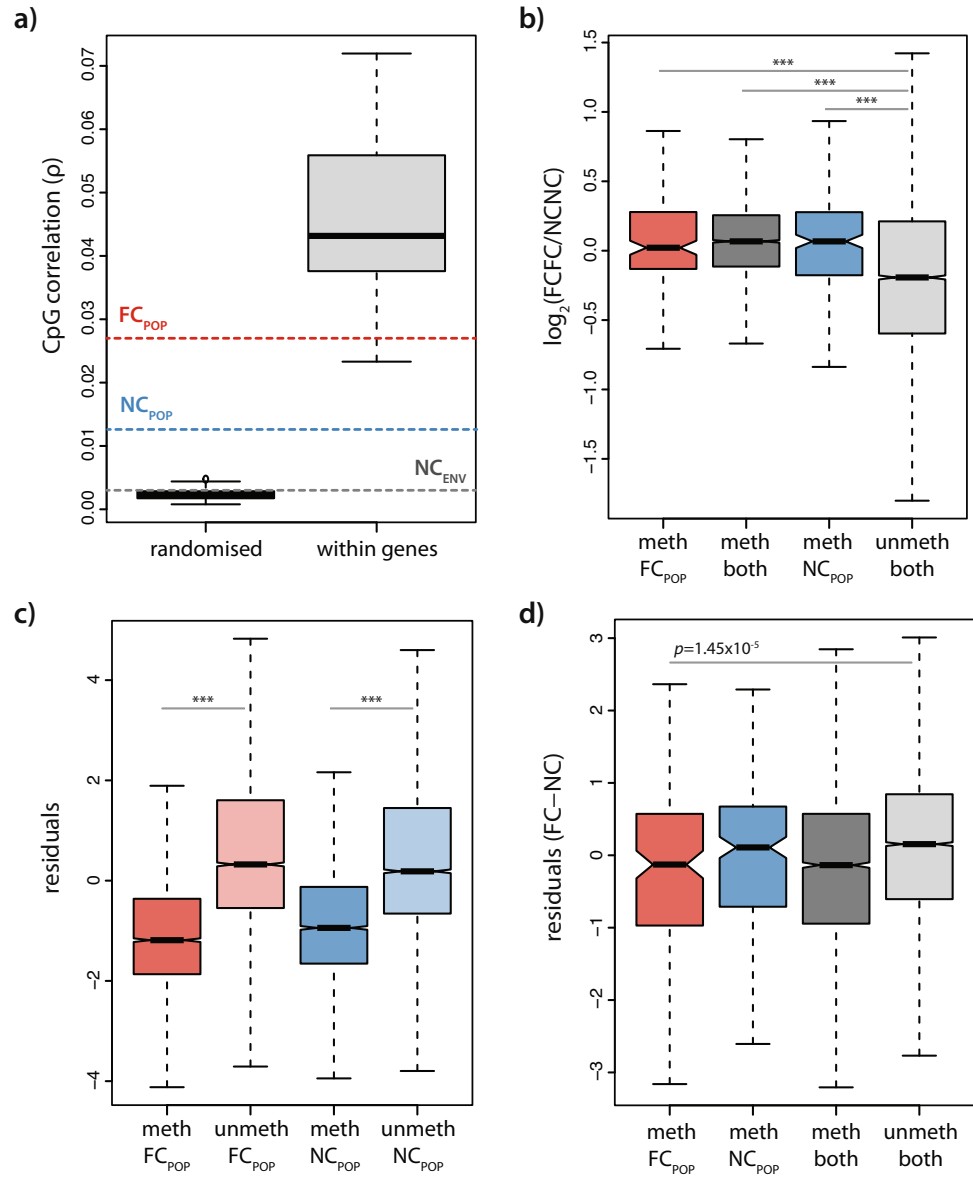

**Fig. 3 | Population-specific methylation is associated with reduced variability in gene expression. a** Boxplot of correlations coefficients between CpGs either within genes or between pairs of CpGs in different genes in random subsamples of methylated genes. Dotted lines represent average correlation coefficient for CpGs within a gene across all environment-specific (grey) and $FC_{POP}$-specific (red; $n = 171$ genes across $n = 3$ biological replicates) and $NC_{POP}$-specific (blue; $n = 217$ genes across $n = 4$ biological replicates) methylated genes. **b** Methylation status of genes increases gene expression. Methylation of genes specific to $NC_{POP}$ or $FC_{POP}$ had similar levels of gene expression (two-sided Wilcoxon test; *** $p < 2.2 \times 10^{-16}$). **c** Gene expression variability (residuals) was significantly lower in methylated genes specific to each population compared to unmethylated genes ($n = 9242$; two-sided Wilcoxon test; *** $p < 2.2 \times 10^{-16}$). **d** Genes that are methylated in $FC_{POP}$ ($p = 1.45 \times 10^{-5}$) but not $N_{POP}$ ($p = 0.45$) have reduced relative expression variability ($FC_{POP} - NC_{POP}$) relative to unmethylated genes (two-sided Wilcoxon test). Data are presented as boxplots. Horizontal lines (bold and black) represent the median. The lower and upper edges of the boxplot represent the interquartile range. Whiskers extend between the highest and the lowest data points within 1.5 times the interquartile range from the median.

fashion within genes, there would be a higher correlation in 5mC levels between CpGs within genes, than between random pairs of CpGs between methylated genes, which we tested by seeking correlations between randomly chosen subsets of genes. We found that correlations between CpGs were higher within methylated genes than *between* methylated genes, indicating that 5mC varies in a coordinated way within genes that are considered methylated (Fig. 3a). Similarly, methylated genes that were unique to each population exhibited a higher correlation between CpGs than would be expected if CpG changes were randomly distributed between genes. However, genes that were unique to a single exposure of a $NC_{ENV}$ did not show these

high correlations between CpG sites within genes. This suggests that evolved changes involve more coordinated changes amongst CpGs within the same genes, whereas short-term environmentally-induced shifts in methylation likely reflects perturbations in individual CpGs that are sparsely distributed across genes (Fig. 3a).

Having established that coordinated changes affected the methylation status of genes, we next tested the consequences of such changes on transcription. As expected, methylated genes tended to have generally higher expression than unmethylated genes ($p < 2.2e$-16, Wilcoxon unpaired test). However, genes that were methylated only in the $FC_{POP}$ were not more highly expressed in $FC_{POP}$ relative to

$NC_{POP}$, and genes methylated only in $NC_{POP}$ were not more highly expressed in $NC_{POP}$. Thus, a change from methylated to unmethylated states was not associated with a change in the expression of those genes (Fig. 3b). This is consistent with our previous analyses showing a lack of association between changes in methylation and gene expression (Fig. 2).

We next asked if methylation could affect transcriptional variability. In other invertebrates, reduced gene expression variability is associated with 5mC in gene bodies[29,53,54], which was also true for this species (Fig. S8). To investigate gene expression variability in our populations we first modelled the relationship between mean expression and the coefficient of variation ($CV^2$) at each gene for each population with a loess function using the samples from our RNA-sequencing analyses (see Methods). We then calculated expression variability for each gene by comparing its $CV^2$ to its predicted $CV^2$ on the basis of its mean expression (residuals; see Methods)[55]. Genes that were methylated in both populations (unique and shared) showed consistently lower gene expression variability than unmethylated genes ($p < 2.2e\text{-}16$, Wilcoxon Unpaired test); Fig. 3c & Fig S9). Genes that were consistently methylated in the $FC_{POP}$ and not in the $NC_{POP}$ showed reduced variability in expression in the $FC_{POP}$ relative to the $NC_{POP}$ ($p < 1.45e\text{-}5$, Wilcox paired test; Fig. 3d). Genes that were methylated in the $NC_{POP}$ but not in the $FC_{POP}$, however, did not show increased variability in $FC_{POP}$ than $NC_{POP}$ ($p = 0.45$, Wilcox paired test). This was because methylated genes in the $NC_{POP}$ showed slightly higher variability in FC than genes that were methylated either specifically in $FC_{POP}$ or in both populations ($p = 0.02$, and $p = 4e\text{-}44$, respectively; Wilcox unpaired test). Overall, these data suggest that methylation reduces variability of gene expression and that these changes in variability were more extreme in the $FC_{POP}$, possibly owing to small increases in the extent of coordination amongst CpGs within those genes (Fig. 3a). Taken together, these data suggest an association between altered gene expression variability and altered tendency for genes to acquire 5mC.

An important question is what caused the differences in 5mC between the two populations. One hypothesis is that genetic differences could be responsible, such as single nucleotide polymorphisms (SNPs) that accumulate between the two populations. However, a very small proportion of genes (1%) that showed altered methylation had highly differentiated SNPs between the populations. Interestingly, genes with 5mC, in general, showed a reduced tendency to contain highly differentiated SNPs compared to genes that were not methylated in either population (Fisher's test, $p = 1e\text{-}15$; Fig. S10). Thus, the majority of differences in the 5mC levels of genes we observed between the populations did not appear to be associated with divergence in DNA sequence.

## Discussion

A role for epigenetic processes in mediating some of the molecular events underlying adaptive divergence is the subject of intense interest and debate[1–4]. How and when such changes are adaptive would depend on the persistence, heritability and function of specific epigenetic modifications during adaptation. We show that DNA methylation and gene expression are each highly sensitive to the loss of parental care, both initially and after 30 generations of exposure. We observed evidence of coordinated changes that accrued over the generations between populations, with multiple CpGs changing within a gene, whereas environmentally induced changes tended to be less coordinated within the gene. Differences in 5mC, either environmental or evolved, were not associated with changes in the levels of gene expression. However, increases in 5mC were associated with reduced gene expression variability, and the magnitude of this response was greater when increases in 5mC were more correlated across CpGs within a gene. These data provide insights into the dynamics of how 5mC evolves in response to a change in the environment and the functions of 5mC within the bodies of transcribed genes.

### Transient and lasting changes in gene expression associated with differences in care

Levels of parental care can shape the development of a number of physiological and behavioural traits that persist throughout the lifespan, including key fitness-related traits such as lifespan, fecundity and survival[46,56]. Conversely, the removal of parental care is associated with disruptions to behavioural development via changes in gene expression across a number of different species[33,44,45,57]. Therefore, it is not surprising that the experience of a $NC_{ENV}$ in burying beetles is the major source of variation in larval gene expression particularly in stress response pathways that disrupt growth and development. Adaptation to the loss of parental care in this species, on the other hand, appears to be associated with the evolution of a stress-tolerant phenotype because we observe[1] reduced responsiveness of genes in the $NC_{POP}$ to the $NC_{ENV}$ (both in terms of expression levels and number of genes induced) and[2] the appearance of a small number of changes that may compensate for stress-related gene expression. This is consistent with the idea that adaptation to stressful environments and/or challenges involves fine-tuning the balance between stress- and growth-related gene expression[58].

We further inspected the genes that were differentially expressed in the $NC_{POP}$ but not $FC_{POP}$ in response to the $NC_{ENV}$ (i.e., new DEGs; Supplementary Data 1). One striking change was the high expression of a cytochrome P450 gene which appears to be a homologue to the Drosophila *Cyp6a20* gene, the deletion of which has been associated with higher levels of aggression and reduced sociality[59]. Also, of particular interest was the differential expression of a number of odorant binding proteins. One such highly expressed gene was a homologue to Drosophila *Obp69a* which is involved in social responsiveness to social experience as well as starvation[60,61]. Changes in the expression of both sets of genes was particularly interesting given that our previous studies have shown that adaptation to the loss of parental care is associated with a greater level of coordination among larvae[50,62] and a shift from sibling conflict to cooperation[51], both of which could aid larvae in locating and utilising the carcass resource in the absence of parents. Our previous whole-genome sequencing analyses of these populations suggest that at least some of the expression changes we observe involve sequence divergence at genes responsible for immunity, stress and behaviour as well as changes in regulatory regions and genes that can control transcription (e.g., transcription factors, co-factors, chromatin modifying enzymes etc)[52].

### Differences in methylation levels associated with differences in care

We sought to determine if any of the observed changes in gene expression (at methylated genes) between our evolving populations could be due to 5mC. 5mC has been shown to be heritable in plants and, to some extent, in other animals[23,26,63,64] – making it a potential candidate for the transmission of evolved epigenetic differences across generations. In rodent model systems, changes in gene expression in developing offspring in response to the loss of parental care have been shown to be, at least in part, mediated by changes in DNA methylation, most notably at promoter regions[24,36,40]. We show that gene body methylation of developing larvae is similarly sensitive to the loss of parental care and 5mC levels can diverge between populations.

Population differences in 5mC could reflect non-genetic inheritance of the epigenetic change, which has been shown to occur for 5mC in plants[23,63,65]. Some of these changes can be derived from environmental sources[64]. Moreover, previous work in *Arabidopsis* has shown that such changes can also arise via spontaneous epimutations that are propagated in stable environments[23]. Alternatively, these

                                                                                                 

changes could be, at least partially, due to cis and/or trans DNA sequence divergence in response to experimental evolution[11,15,17,18,35]. For example, in sticklebacks, some but not all population-level variation in 5mC has been linked to divergence at *cis* or *trans*-acting loci[15,17]. In contrast, the number of differentially methylated regions between species of Darwin's finches was correlated with phylogenetic distance but did not overlap with genetic differences[18]. Similarly, in our study, SNPs with large differences in allele frequencies between the populations, rarely overlapped with genes showing altered 5mC, suggesting that at least *cis*-acting DNA sequence change is unlikely to explain the majority of the differences we found in methylation. This is also consistent with our previous work using whole-genome sequencing, which showed significant genetic divergence between populations that did not overlap with any of the methylated genes described here[52]. These differences may reflect variation in 5mC systems and DNMT evolution across different lineages, which cause species-specific differences in where and how changes in methylation might accumulate[24,30,42,66,67].

To what extent might population differences in 5mC be part of the process of adapting to different environments? There was no tendency for genes showing differential 5mC to be associated with changes in levels of gene expression between populations, suggesting that gene expression and DNA methylation are separately evolving processes. This is consistent with a growing body of work showing a lack of association between differences in invertebrate methylation (5mC that is primarily in gene bodies) and differences in gene expression across a number of different qualitatively different environmental exposures[29,33–36]. Though it is possible that any regulatory outcomes of 5mC on transcription depend on its positioning relative to nucleosomes and/or the combinatorial actions of other neighbouring epigenetic marks[29,68,69]. It is also possible that other more widespread epigenetic marks, such as specific chromatin modifications, could evolve and contribute to the genome-wide gene expression changes we observe here, both at methylated and unmethylated genes[70,71]. We did, however, observe a tendency for the variability of gene expression to change in association with changes in methylation. Genes that were methylated tended to have reduced variability. Previous work has documented a clear association between methylation and lower variability in gene expression in a number of arthropod species[29,53,54]; our study further suggests that this relationship may have a causal basis. Interestingly, the subset of genes that possesses gene body methylation has been highly conserved across invertebrate evolution, and the same orthologous genes tend to be methylated across most species[29]. However, there do appear to be subtle differences among species[29,35] and it would be interesting to investigate whether genes that lose and/or gain methylation specifically in one species similarly show changes in expression variability.

Could the reduced variability associated with gain of methylation have a function in adapting to presence or absence of parental care? Interestingly, the tendency of methylated genes to show decreased variability was particularly pronounced in the FC population. This might reflect part of a key feature in the response to parental care. In caring for offspring, parents create benign nutrient-rich environments for their young, and in doing so can mask the accumulation of genetic of variation amongst individuals, a long-term consequence of which is reduced phenotypic variability within populations[52,72,73]. Therefore, the loss of expression variability observed here, in the presence of care, might be driven by the persistent exposure to abundant diets and/or stable environments[4] and key housekeeping and growth-responsive genes accumulating 5mC. On the other hand, the loss of care (both evolved and environmental), led to more sparse changes in 5mC, an effect that might be mediated by disruptions to DNMT1 activity in response to cellular stress[7]. Although these changes were small, it suggests that genes might acquire and lose methylation at different rates depending on whether the environmental conditions incurred were benign or stressful. This may explain why 5mC in genes uniquely

methylated in the NC population were not as coordinated within genes and exhibited a weaker association with expression variability compared to the FC population. These changes could have long-term consequences for the canalisation of gene expression networks in response to the loss of care[74] as well as for fitness across the lifespan. For example, both increased transcriptional variability and methylation loss could underpin higher rates of aging in the absence of parental care and/or other stressors[75–77]. Changes in methylation might either directly provoke changes in variability or be a consequence of the transcriptional environment induced when care is present or absent. Moreover, the use of CRISPR-cas9 and RNAi mediated knockdown of insect DNMTs has revealed potentially interesting links between DNMT1 and reproduction in insects, functions that appear to be independent of 5mC[38–41]. It would, therefore, be interesting to test whether deletion of DNMTs affects the fitness benefits of parental care across the lifespan via the inappropriate loss of methylation and increased transcriptional variability.

In sum, we have found that most of the changes in 5mC levels we detected were predominantly environmentally induced and persisted only in the short-term. However, the epigenetic differences that accrued in 5mC between the two types of experimental populations were likely distinct from cis-acting DNA sequence variation and were related to functional changes in the variability of gene expression rather than levels of gene expression per se. This suggests that population differences in gene body 5mC might reflect gene expression variability within populations, which is known to be critical parameter in the evolution of resilience to stressful environments[58,78]. Taken together, these data provide insights into the interplay between environmental variation, the evolution of 5mC in gene bodies and functional outcomes in invertebrates.

## Methods
### Breeding design & Experimental Evolution
We analysed experimental populations of *Nicrophorus vespilloides* that had been evolving under different regimes of parental care, and which were founded from a genetically diverse population generated by interbreeding beetles from multiple wild populations across Cambridgeshire. These populations are described in detail in Schrader et al. [50]. and comprise a total of 4 populations, two blocks (Block 1 and Block 2; separated by 1 week) containing two populations evolving with (FC_POP) or without parental care (NC_POP). On the 29th generation, seventeen days after their emergence as adults, when individuals were sexually mature, we paired 15 males and females within each population ($n = 30$ pairs in total). Each pair was placed in a separate breeding box with moist soil and a thawed carcass (10-12 g). We then placed each breeding box in a cupboard, and allowed parents to prepare the carcass and for the female to lay the clutch of eggs. After 53 h, populations were split such that both parents were either removed (in keeping with the procedure experienced by the NC_POP) or left in the breeding box. This produced offspring from both populations in their evolved condition (FC_POPFC_ENV & NC_POPNC_ENV) as well as their reciprocal environments (FC_POPNC_ENV, NC_POPFC_ENV). At 80 h post-pairing (approx. 10-12 h post-hatching), first-instar larvae were collected from surviving families. Only families with brood sizes between 20-30 larvae were used for subsequent RNA-sequencing analyses.

### Larval tissue dissection, RNA extraction & sequencing
For each family, RNA from four heads of first instar larvae was pooled and extracted using Trizol (Invitrogen). The resulting sample group sizes for RNA-sequencing analysis were: FC_POPFC_ENV ($n = 12$), FC_POPNC_ENV ($n = 11$), NC_POPNC_ENV ($n = 12$), NC_POPFC_ENV ($n = 12$) with both replicated blocks being represented. Total RNA quality was checked using the BioAnalyzer System (Agilent), and yield was quantified using a Qubit RNA Assay Kit (Thermo Fisher). PolyA-selected RNA libraries were constructed and sequenced (150 bp paired-end) at a

depth of 30x by Novogene (Hong Kong). The resulting sample group sizes for gene expression analysis were: 5-6 libraries for each group within each block (total of 46 libraries).

## RNA Sequencing & read mapping

Reads were trimmed using TrimGalore (0.5.0; https://github.com/FelixKrueger/TrimGalore) to remove adaptor sequences, perform quality trimming and discard low-quality reads. Reads were mapped and quantified using a custom pipeline using HiSat2 (2.1.0) and Stringtie (2.0.3)[79]. Transcript abundance estimation was based on counting reads aligned to the *N. vespilloides* reference transcriptome (NCBI Refseq Assembly: GCF_001412225.1)[43]. Lowly expressed genes (those with less than 15 counts in more than 90% of samples) were filtered from raw counts table leaving a total of 12,772 expressed genes in the dataset to be analysed. All subsequent post-processing and statistical analyses were performed in *R* version 4.1.2[80]. Data wrangling and visualisations in *R* were performed using the tidyverse suite[81].

## DAPC using adegnet and MCMCglmm

To compare overall patterns of gene expression plasticity, we performed a discriminant analysis of principal components (DAPC), which linearises principal components within a dataset into a single discriminant function, enabling the comparison of overall levels of variation between groups. Gene counts were normalised and log-transformed using a regularised log transform in DESeq2 (version 1.26.0)[82] and a discriminant function was built by defining each population's response to their reciprocal environments using adegenet (version 2.1.3)[83]. To compare differences in gene expression plasticity between populations we used a generalised linear model (GLM) using the MCMCglmm package (version 2.29)[84] to model DAPC score as a function of population background and current environment (and an interaction between the two) using a default prior that assumes a normal posterior distribution with large variances for the fixed effects and a weakly informative (flat) prior.

## Differential expression analysis

Differential expression was analysed using DESeq2 using custom scripts in R. Log2-fold change estimates for each differentially expressed gene (DEG) were shrunk to generate more conservative estimates of effect size using the ashr shrinkage estimator[85]. Moreover, to focus on genes that are differentially expressed at higher thresholds we considered a gene to be significantly differentially expressed only if it had increased by a 2-fold change (i.e., Log2FoldChange >= 1). We included the separate blocks as a covariate in all the analyses to increase power but also to account for minor fluctuations in gene expression across the blocks of replicate populations.

## Analysis of gene expression and gene expression variability

To extract mean and variability for each gene we extracted normalised counts for each RNA-sequencing sample using DESeq2 according to previous methods[55]. First, the coefficient of variation was calculated using the equation: $CV^2 = \sigma^2/\mu^2$. The coefficient of variation was then plotted against the mean for each gene and fitted with a smoothed local regression using the locally estimated scatterplot smoothing algorithm (loess) for each population in R. To quantify the extent of variability at each gene we extracted residuals from the model fit, where high deviations from the fitted model were indicative of increased variability. The variability of gene expression for different sets of methylated and unmethylated genes, as well as the mean expression of these sets in different populations, was then compared statistically using the Wilcoxon unpaired test.

## Functional annotation

Functional enrichment analyses were conducted using the topGO R package (version 2.38.1)[86] to identify over-representation of particular

functional groups within the DEGs in response to the removal of care as well as the evolved response to the removal of care, based on GO classifications using Fisher's exact test. GO terms were annotated to the *N. vespilloides* genome using the BLAST2GO (version 5.1.1)[87] workflow to assign homologues to the *Drosophila* non-redundant protein databases[88]. To increase functional predictions of *N. vespilloides* genes these annotations were supplemented with GO term assignments based on ortholog searches within Arthropods using eggNOG-mapper[89] and OrthoDB (v10.1)[90].

## Bisulfite sequencing & read mapping

For each family, DNA from heads of first instar larvae was pooled and extracted using the Qiagen DNEasy Blood & Tissues kit (Qiagen). Total DNA quality was checked using the BioAnalyzer System (Agilent), and yield was quantified using a Qubit DNA HS Assay Kit (Thermo Fisher). DNA methylation libraries were constructed and sequenced (150 bp paired-end) at a depth of 30x by Novogene (Hong Kong). The resulting sample group sizes for methylation analysis were: 3-4 libraries for each group (pooled from the same families used for RNA-sequencing): FC_POPFC_ENV, FC_POPNC_ENV, NC_POPNC_ENV, NC_POPFC_ENV (*n* = 3-4/group made up of 8-10 larvae from 4-5 distinct families per library pool; total of 14 libraries). Following initial QC (using FastQC v0.11.9), reads were trimmed using TrimGalore (0.6.4) to remove adaptor sequences and poor-quality reads. Bisulfite conversion efficiency across all samples was estimated by Novogene and was between 99.14-99.50% for all samples. Bisulfite-converted reads obtained from each library were mapped to the *N. vespilloides* genome using Bismark (v0.22.3)[91] and were quantified following deduplication of reads. Mapping rates for samples to the *N. vespilloides* reference genome were 65.45% ± 5.48, which after deduplication gave an average coverage of 17.78x ± 1.79 (mean ± SD). Only CpGs with at least 10 reads were retained for subsequent analyses.

## Differential methylation analysis

For differential methylation analysis of individual CpGs, we first tested each site in every sample to determine sites that were significantly methylated using a binomial test. We then tested for differential methylation across sites using a weighted binomial glm (*p*-values were adjusted using Benjamini-Hochperg corrections for multiple testing). We then mapped CpGs to their feature using our custom genome annotation (see below for details) and bedtools (v2.29.2)[92]. For analyses of gene body methylation, we collapsed across all CpG sites within each genic region (using bedtools) to compute the average weighted methylation value for each gene for each sample. Genes were clustered into methylated and unmethylated clusters using mixture models using the mixtools R package (v2.0.0). To determine which genes were differentially methylated we used a weighted binomial GLM (base R stats package) and adjusted for multiple testing using the Benjamini-Hochberg procedure. Genes were considered differentially methylated if they had an adjusted *p* < 0.01. Fisher's Exact tests for enrichment, to test for significant associations (either hypo- or hyper-enrichment), were run using the base *R* stats package.

## Identification of methylated genes

We estimated the propensity of genes to acquire or lose methylation (i.e. the tendency of a gene to be methylated or unmethylated) similarly to previous studies[29]. CpGs were classified as either methylated or unmethylated by using a kmeans clustering algorithm with two states, transforming the percentage methylation at each CpG into either 1 (methylated) or 0 (unmethylated). The methylation of each CpG was then used as the input for a generalised linear model with the gene that the CpG was assigned to as a random effect implemented with the lme4 (v1.1-34) package[93] in *R*. We then extracted the random intercept of each gene using the 'ranef' function, which revealed a clear bimodal distribution of genes that did and did not possess methylation

(Fig. S7). We considered a gene to be methylated if the random effect was greater than 2.5 for each population to help identify sets of genes that were methylated in each population.

## Correlation of 5mC across CpGs within and between genes

We selected genes that were found to be methylated in both populations according to the methods in the previous section. Within these genes we selected 100 genes at random and extracted all pairs of CpGs using the function 'expand.grid' in *R*, and repeated this 100 times to obtain an empirical estimate of the Pearson correlation between all pairs of CpG within the same gene. We then shuffled the methylation of CpGs within the set and performed the same analysis to obtain a distribution of the correlation in CpGs between different genes. We also calculated the correlation between all pairs of CpG within the same gene in the subsets of methylated genes that showed population-specific differences between the populations (both $NC_{POP}$ and $FC_{POP}$-specific) and genes that showed methylation states that were unique to a single exposure of a $NC_{ENV}$ (environmental-specific) to show how these values deviated from our subsampling procedure.

## SNP extraction from bisulfite sequencing reads

We used BISCUIT (v1.0.0) to extract SNPs from bisulfite sequencing reads (https://github.com/huishenlab/biscuit). Briefly, we used mapped reads to call SNPs using BISCUIT defaults. The resulting VCF was filtered by 2x the max coverage (for each sample) and converted to a BED file. We used the bedtools (v2.29.2)[92] 'intersect' function to extract all SNPs within genes and considered a gene to have DNA sequence divergence if SNPs falling within that gene were present at a frequency greater than 0.8 in one population but less than 0.2 in the other.

## Genome annotation

To annotate exons in each genome we used existing annotations, excluding genes that were split across multiple contigs. To annotate regions which may contain promoters or enhancers (5′ UTR), we took 1,000 bases upstream of each gene, excluding genes where this exceeded the contig start or end point. We annotated introns based on the position of exons, excluding genes that were split across multiple contigs. To annotate TEs, we used RepeatModeller (v2.0.1) to generate a model of TEs for the *N. vespilloides* genome, and then RepeatMasker (v4.1.0) to annotate TEs based on the model for that genome[94].

## Reporting summary

Further information on research design is available in the Nature Portfolio Reporting Summary linked to this article.

# Data availability

All raw sequencing data generated in this study have been submitted to the NCBI Gene Expression Omnibus (GEO) under accession number (GSE171776). Source data are provided as a Source Data file. Source data are provided with this paper.

# Code availability

All code for the analyses contained within this paper can be found at: https://github.com/r-mashoodh/nves_MethEvol.

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

## Acknowledgements

We would like to thank Benjamin Jarrett, Sue Aspinall, Darren Rebar, Ana Duarte and Matt Schrader for technical assistance with maintaining burying beetle populations. We would also like to thank Syuan-Jyan Sun and Samuel Lewis for technical assistance with the molecular work. A special thanks to Benjamin Jarrett and Chris Cunningham for feedback and many helpful discussions. This project was supported by the European Research Council (310785 BALDWINIAN_BEETLES) and a Royal Society Wolfson Merit Award, both to R.M.K. The molecular work was supported by a BBSRC Future Leaders Fellowship (BB/R01115X/1) and a Royal Society Small Research Grant (RGS\R1\191162) to R.M.

## Author contributions

R.M. and R.M.K. conceived the study. R.M. conducted the experimental work. R.M., J.W. and P.S. performed formal analysis of the data. R.M. and P.S. performed data interpretation and visualisation. R.M. and P.S. wrote the manuscript. All authors discussed the results and contributed to the final manuscript.

## Competing interests

The authors declare no competing interests.
