## [Peer Review File · Nature Communications]

REVIEWER COMMENTS

Reviewer #1 (Remarks to the Author):

Remarks

I liked the experiment and its population seem very interesting. I did wonder if some of the effects are less certain that presented. I think the paper will be of great interest to NC readership.

Title. Where there not more unique methylated genes for the FCpop than the NCpop? Title suggests the opposite.

L18. Stochastic processes are suggested as a possibility. But this idea is never really returned to. But I feel the data might be able to give some estimate of it. How many genes were methylated in only one of the four lines or in one sample? It seems a reasonable suggest that the data can hint at so why not try?

L83. Why is there any expectation of methylation influence relative gene expression differences? It has been looked at in this species and in many other insects. In all but one case, jewel wasp eggs, there is no association. Maybe the author means if their experiment produced a new phenomenon?

L101. The study is contrasting populations and the questions are framed at the population level. Why is the analysis not also at the population level. It would greatly reduce the statistical power, some of which could be recouped by a nested analysis, but that is really a strong reason. Can this design ever go past two biological replicates per treatment? I see samples from the same population as technical, not biological, replicates.

L260. But those are still incredibly low correlation coefficients.

L290. What are the actual p-values associated with these tests for fig 3d? That looks like a very, very small difference that could easily be driven by a power curve difference due the number of uniquely methylated genes of each population 171 vs 217. Although that would be in the opposite direction as expected for unique NCpop methylated genes.

L294. Maybe. That is very certain statement to attributed to a very small difference of correlation and a very small difference in gene expression variation. Normally that is just writing but close to half the first paragraph of the discussion is also dedicated to this association.

L315. They are fractionally more coordinated. A difference of correlation of about 1% absolutely is not a huge difference. Also, the evolved NCpop showed a lower correlation suggesting a loss of coordination under artificial selection. This is not a strong result but it is very intriguing.

L361. It is of no value to reference mammalian methylation. Over many years now the field has established that insects are fundamentally different than mammals. Mammals should not be compared to insects nor should they provide any basis for the expectation of what the association between methylation and gene expression should be within insects.

Schmitz et al. 2019. DNA Methylation: Shared and Divergent Features across Eukaryotes. Trends Genetics 10.1016/j.tig.2019.07.007 .

Duncan et al. 2022. Phenotypic Plasticity: What Has DNA Methylation Got to Do with It? Insects <https://doi.org/10.3390/insects13020110>

Bogan. 2024. Potential role of DNA methylation as a driver of plastic responses to the environment across cells, organisms, and populations. Genome Biology and Evolution

L366. Again, why are mammals being referenced? Insects are different and that has been known for many years. We do not have to state again they are different as a discussed result of a paper.

L401. Uniquely methylated gene of the NCpop did not show reduced gene expression. That directly works against this statement. It's one of the more interesting results. It is a very consistent association among insects and your evolved population do not show it for the de novo methylated genes.

This comment bleeds into the next paragraph.

L409. But the NCpop is the one under selection pressure for the loss of parental care. The FCpop is acting under its evolved environments and is likely just operating under drift.

L412. But in order not to see gene expression variation due to sequence differences then those loci would need to be not be expressed, no? So parental care keeps off genes that have more genetic variation? That can be tested here. Do DEGs under No Care show higher than average SNP density? 5mC promotes mutation though because it is associated with open chromatin; Gladstad et al. 2016. Effects of DNA Methylation and Chromatin State on Rates of Molecular Evolution in Insects. G3.

L430. DNMT1 link to reproduction was known long before Kronauer's CRISPR KD. It is not a recent phenotypic association. Zweir et al. 2012. DNA methylation plays a crucial role during early *Nasonia* development. *Insect Molecular Biology*, 21, 129–138.

Editorial Suggestions/Questions

Abstract – “of a biparental”

L29. Is there any insect that approaches the level of the lowest mammalian genome methylation level?

L92. Should the second reciprocal environment not be NCpopFCenv?

L189. Is 50% “highly similar?”

L580. What correlation was used?

This reference might also explain why some methylated gene have higher expression and lower variation: Kucharski et al. 2023. The PWWP domain and the evolution of unique DNA methylation toolkits in Hymenoptera. *iScience* 10.1016/j.isci.2023.108193.

Reviewer #2 (Remarks to the Author):

General comments

I liked this manuscript very much, and my remarks are only suggestions, mostly trivial.

The main conclusion, that long-term selection can change methylation pattern, and reduce variance in gene expression but not absolute gene expression is novel and noteworthy. It will be of interest to

everyone who works on epigenetics and evolution in invertebrate animals. It is less relevant to vertebrates and plants whose methylation patterns are different.

As far as I can tell, the data have been thoroughly and appropriately analysed. The experimental design is sound and properly replicated. The methods are clear.

I thought that the manuscript could start by more broadly discussing the role of phenotypic plasticity and epigenetics evolution, by asking the questions that many are asking, but not satisfactorily answering. Does selection change gene expression rather than gene sequence? Is the methylation pattern replicated along with DNA, or is it re-established in offspring? Is an evolved change in DNA methylation pattern associated with a loss of phenotypic plasticity?

The study is germane to the hypothesis of phenotype-first evolution. Does, as championed by West-Eberhard (1989), Oldroyd (2023), Jablonka and Lamb (2014), the methylome and gene expression response to environmental change lead long-term change in gene expression and a temporary loss of phenotypic plasticity?

I guess what I'm suggesting is a small pivot away from the mechanistic focus to a more evolutionary one. Just my bias and interest.

Specific comments

Abstract

At least in insects it isn't clear that epigenetic modifications are transmitted between generations. Instead, patterns of methylation may be conserved across generations (i.e re-established in offspring). You say this in the main paper, but I think it needs to be in the abstract too.

L 9 Should this be 'incipient species'?

L 21 Delete 'are going to'.

L 33 Maybe say 'evolutionarily' conserved, or 'phylogenetically' conserved. As written it might be construed in the sense of heritable.

L 63 Should read 'hatched'.

L 129 Please define OR at this first use.

L 197 Two probabilities of differential expression. Which is correct?

L 198 It seems strange that a 2 percentage point difference in the proportion of DMG that also show differential gene expression could result in such a wildly different statistical association. I assume that this is due to the sample size. So please state the number of DMG analysed at lines 197 and 198. Also, I don't understand how an association can be highly significant, while simultaneously being at 'chance levels'.

L221 The p-value in the evolved response is extremely low, yet you say that there is no association. Please provide a few words to explain what this test is doing.

L 231 What is a 'fixed evolved difference'? A permanent change in methylation pattern? Do you really want to say 'fixed'?

L352 Delete _

L361 Not sure what 'stable' means in this context. Across generations? Presence/absence is similar across individuals?

L370 I think it would be clearer to simply say 'inheritance of the epigenetic change'.

L 399. Please provide references to this previous work.

L 457. 'Each pair was placed in a separate breeding box.' Is repeated in the next sentence.

JABLONKA, E. & LAMB, M. J. 2014. Evolution in four dimensions, Cambridge MA USA, Bradford.

OLDROYD, B. P. 2023. Beyond DNA. How epigenetics is changing the way we think about evolution, Melbourne, Melbourne University Press.

WEST-EBERHARD, M. J. 1989. Phenotypic plasticity and the origins of diversity. Annual Review of Ecology and Systematics, 20, 249-278.

RESPONSE TO REVIEWERS

We thank the reviewers for their helpful comments and helping us improve our manuscript.

Below we address each point in turn. Changes within the final manuscript are highlighted in red and our responses to each reviewer comment are in blue.

Reviewer #1 (Remarks to the Author):

Remarks

I liked the experiment and its population seem very interesting. I did wonder if some of the effects are less certain that presented. I think the paper will be of great interest to NC readership.

Title. Where there not more unique methylated genes for the FCpop than the NCpop? Title suggests the opposite.

Methylation is changing in different ways across both populations (both in terms of levels as well as uniquely methylated genes) – the title is agnostic to this.

L18. Stochastic processes are suggested as a possibility. But this idea is never really returned to. But I feel the data might be able to give some estimate of it. How many genes were methylated in only one of the four lines or in one sample? It seems a reasonable suggest that the data can hint at so why not try?

In this manuscript, by stochastic we mean differences in methylation that occur independent of environmental differences, as opposed to changes in methylation that occur due to responses to the environment (now clarified in the revised version, line 21-23). Both of these types of differences may occur in the different populations and our experimental design does not enable us to distinguish between them. To definitively identify stochastic changes in methylation we would have to perform the experimental evolution experiments with orders of magnitude more independent replicate populations for each condition measured at every generation, which, for methylation, has only been performed for Arabidopsis (now referenced in the discussion), but which, in our experimental system, would require several more years to complete. This is why our study focuses on the largest changes common to all biological replicates.

L83. Why is there any expectation of methylation influence relative gene expression differences? It has been looked at in this species and in many other insects. In all but one case, jewel wasp eggs, there is no association. Maybe the author means if their experiment produced a new phenomenon?

In arthropods there is clearly a strong association between the propensity of genes to be methylated and their expression level -- many studies have demonstrated that methylated genes are depleted for both highly expressed and lowly expressed genes. Moreover, genes with consistent expression across tissues are more likely to be methylated (de Mendoza *et al.*, 2020 *JMB*; Lewis *et al.*, 2020 *PLoS Genetics*; Schmitz *et al.*, 2019 etc). Few studies have investigated whether these associations reflect causal relationships. Using experimental evolution we can test whether changes in methylation (presence vs absence) are associated with changes in either expression levels or variability in expression, which we believe is an important addition to what is known so far from static gene expression vs methylation level correlations. We have edited the Introduction to clarify what we are measuring (lines 89-90).

L101. The study is contrasting populations and the questions are framed at the population level. Why is the analysis not also at the population level. It would greatly reduce the statistical power, some of which could be recouped by a nested analysis, but that is really a strong reason. Can this design ever go past two biological replicates per treatment? I see samples from the same population as technical, not biological, replicates.

The biological replicates contain different individuals derived from different families which were originally derived from a wild population. Thus, the replicates are biological and not technical.

We agree that more replicate populations would give more power, and as the reviewer points out earlier (comment L18), could allow us to estimate stochastic change. However, this is a major undertaking and requires substantial funding, which we are currently working on. The current study is already 3+ years of animal work alone and the results already highlight the novelty and importance our manuscript.

L260. But those are still incredibly low correlation coefficients.

We now acknowledge this in the text.

L290. What are the actual p-values associated with these tests for fig 3d? That looks like a very, very small difference that could easily be driven by a power curve difference due the number of uniquely methylated genes of each population 171 vs 217. Although that would be in the opposite direction as expected for unique NCpop methylated genes.

Thank you for raising this point. To investigate whether differences in sample number could be responsible we downsampled (100 times) to the smaller number of genes. In each of these iterations the variability was lower in the FC population than the NC population, which suggests that differences in the size of the two gene sets is unlikely to be responsible for the statistical difference we observe (see Figure below).

We have also added the actual p-values (related to Figure 3d) to the Results section (lines 296-302).

L294. Maybe. That is very certain statement to attributed to a very small difference of correlation and a very small difference in gene expression variation. Normally that is just writing but close to half the first paragraph of the discussion is also dedicated to this association.

We have edited the language here to tone this statement down (now lines 303-306)

L315. They are fractionally more coordinated. A difference of correlation of about 1% absolutely is not a huge difference. Also, the evolved NCpop showed a lower correlation suggesting a loss of coordination under artificial selection. This is not a strong result but it is very intriguing.

We agree this is a very intriguing result and warrants mention. We have edited the language to tone these statements down (lines 303-306, 324-326 and 429-430).

L361. It is of no value to reference mammalian methylation. Over many years now the field has established that insects are fundamentally different than mammals. Mammals should not be compared to insects nor should they provide any basis for the expectation of what the association between methylation and gene expression should be within insects.

Schmitz et al. 2019. DNA Methylation: Shared and Divergent Features across Eukaryotes. Trends Genetics 10.1016/j.tig.2019.07.007 .

Duncan et al. 2022. Phenotypic Plasticity: What Has DNA Methylation Got to Do with It? Insects <https://doi.org/10.3390/insects13020110>

Bogan. 2024. Potential role of DNA methylation as a driver of plastic responses to the environment across cells, organisms, and populations. Genome Biology and Evolution

We agree, however, there is a subset of the broad Nature Communications readership (mammalian epigeneticists, ecologists etc) where this point might not be readily apparent and so we believe it is important to make these distinctions clear.

L366. Again, why are mammals being referenced? Insects are different and that has been known for many years. We do not have to state again they are different as a discussed result of a paper.

We have removed this sentence.

L401. Uniquely methylated gene of the NCpop did not show reduced gene expression. That directly works against this statement. It's one of the more interesting results. It is a very consistent association among insects and your evolved population do not show it for the de novo methylated genes.

This comment bleeds into the next paragraph.

There are still reductions in variation of uniquely methylated genes in both populations (Figure 3c). The effect on variability is stronger relatively in the FC population (now made more clear in the Results lines 294-296 and with an additional Fig S9). As we point out in the Discussion this could be a function of the contrasting response to parental care compared to the stress of losing care, both of which are uniquely complex and multifaceted environments.

L409. But the NCpop is the one under selection pressure for the loss of parental care. The FCpop is acting under its evolved environments and is likely just operating under drift.

The beetles used in this study were derived from wild populations. In nature, there is biparental care but this species is also facultative in its expression of care (sometimes they provision care, sometimes they don't; see lines 58-66) and so, in this experiment, the artificial conditions are forcing care (Full Care) just as much as we are experimentally removing it (No Care). Therefore, neither environment can be considered 'natural'. What is interesting is that this creates contrasting patterns of methylation and effects on transcriptional processes – which has implications for how we interpret the consequences of DNA methylation and its response under contrasting environments.

L412. But in order not to see gene expression variation due to sequence differences then those loci would need to be not be expressed, no? So parental care keeps off genes that have more genetic variation? That can be tested here. Do DEGs under No Care show higher than average SNP density? 5mC promotes mutation though because it is associated with open chromatin; Gladstad et al. 2016. Effects of DNA Methylation and Chromatin State on Rates of Molecular Evolution in Insects. G3.

We have previously measured standing genetic variation (i.e., genetic polymorphism) using population genetic metrics (π and/or Watterson's θ). There is no reason to expect that variability in SNPs (standing genetic variation) would be more extreme in the No Care

population. In fact, there should be more genetic variability in the Full Care population relative to the No Care population, which is what we showed using whole-genome sequencing (Mashoodh et al., 2023 *Evolution Letters*). This suggests a global loss of genetic variation and strong directional (purifying) selection in response to the loss of care and is consistent with adaptation from standing genetic variation. The relationship between parental care, standing genetic variation and gene expression is definitely an interesting question but beyond the scope of this study and requires partitioning individual-, family- and population-wise variation on gene expression with a high number of individuals (which we cannot do within the current experimental design).

The focus of the SNP analysis in the current manuscript is to see if our observed patterns of methylation can be attributed to allele frequency differences at SNPs between populations (see Methods; lines 609-612). We show (Fig S10) that differential SNPs cannot account for genes becoming uniquely methylated in the FC and NC population. These are SNPs that show high frequency differences between populations at those genes. Moreover, there are overall very few extreme SNPs in general within these genes. This is consistent with our previous work, which showed signatures of selection at a number of candidate gene loci (referenced on lines 383-387). Interestingly, none of these genes are considered methylated genes (genes that possess methylation). This suggests that genetic differences between populations are unlikely to account for the methylation differences we observe. This does not mean that mutation at open chromatin or methylated CpGs do not occur – they just might not occur on the timescales of adaptive evolution that we are examining here.

We have rewritten parts of the manuscript to be more clear about what we are measuring -- both in the description of the Results (now lines 310-314) and the Discussion (now lines 377-387).

L430. DNMT1 link to reproduction was known long before Kronauer's CRISPR KD. It is not a recent phenotypic association. Zweir et al. 2012. DNA methylation plays a crucial role during early *Nasonia* development. *Insect Molecular Biology*, 21, 129–138.

We have edited this sentence and cited this paper as well as others to make this point better (now lines 440-442).

Editorial Suggestions/Questions

Abstract – “of a biparental”

Fixed

L29. Is there any insect that approaches the level of the lowest mammalian genome methylation level?

Not that we are aware of. *S. maritima* (centipede) is ~30% but most mammals are greater than 60%.

L92. Should the second reciprocal environment not be NCpopFCenv?

Typo has been fixed.

L189. Is 50% “highly similar?”

We have edited this line for clarity (line 195).

L580. What correlation was used?

‘Pearson’ has been added to the text (line 596).

This reference might also explain why some methylated gene have higher expression and lower variation: Kucharski et al. 2023. The PWWP domain and the evolution of unique DNA methylation toolkits in Hymenoptera. *iScience* 10.1016/j.isci.2023.108193.

Thanks for the valuable feedback.

Reviewer #2 (Remarks to the Author):

General comments

I liked this manuscript very much, and my remarks are only suggestions, mostly trivial.

The main conclusion, that long-term selection can change methylation pattern, and reduce variance in gene expression but not absolute gene expression is novel and noteworthy. It will be of interest to everyone who works on epigenetics and evolution in invertebrate animals. It is less relevant to vertebrates and plants whose methylation patterns are different.

As far as I can tell, the data have been thoroughly and appropriately analysed. The experimental design is sound and properly replicated. The methods are clear.

I thought that the manuscript could start by more broadly discussing the role of phenotypic plasticity and epigenetics evolution, by asking the questions that many are asking, but not satisfactorily answering. Does selection change gene expression rather than gene sequence? Is the methylation pattern replicated along with DNA, or is it re-established in offspring? Is an evolved change in DNA methylation pattern associated with a loss of phenotypic plasticity?

The study is germane to the hypothesis of phenotype-first evolution. Does, as championed by West-Eberhard (1989), Oldroyd (2023), Jablonka and Lamb (2014), the methylome and gene

expression response to environmental change lead long-term change in gene expression and a temporary loss of phenotypic plasticity?

I guess what I'm suggesting is a small pivot away from the mechanistic focus to a more evolutionary one. Just my bias and interest.

Thanks for your supportive and helpful comments.

We have now edited the introduction to make reference to these hypotheses and perspectives (now lines 8-10 & 14-15). These are complex ideas and are surrounded by much debate about the definition and measurement of true 'plasticity' – which we did not test explicitly here.

However, we believe that the plausibility of each of these hypotheses are going to depend heavily on how and when epigenetic mechanisms exert their actions, which are going to vary by species and the epigenetic mark in question (lines 23-26). We use our study to ask questions about the role of epigenetic mechanisms in adaptive evolution as well as how evolutionary studies can highlight how and when these mechanisms do and do not operate. This is why we think focusing on mechanism is so important and why our results are exciting in the context of this field.

Specific comments

Abstract

At least in insects it isn't clear that epigenetic modifications are transmitted between generations. Instead, patterns of methylation may be conserved across generations (i.e re-established in offspring). You say this in the main paper, but I think it needs to be in the abstract too.

We have edited the abstract to reflect this.

L 9 Should this be 'incipient species'?

We have added 'closely-related' to this line.

L 21 Delete 'are going to'.

Done.

L 33 Maybe say 'evolutionarily' conserved, or 'phylogenetically' conserved. As written it might be construed in the sense of heritable.

Heritability is the right word here. The study cited here showed that offspring methylation is more similar to parental methylation across bee families. Moreover, specific interindividual

patterns of paternal 5mC are recapitulated even when offspring are sired using *in vitro* fertilisation. This suggests some degree of transmission and/or heritability from parent to offspring (and a lack of DNA methylation reprogramming).

L 63 Should read 'hatched'.

Fixed.

L 129 Please define OR at this first use.

Inserted 'odds-ratio' here.

L 197 Two probabilities of differential expression. Which is correct?

Typo has been removed.

L 198 It seems strange that a 2 percentage point difference in the proportion of DMG that also show differential gene expression could result in such a wildly different statistical association. I assume that this is due to the sample size. So please state the number of DMG analysed at lines 197 and 198. Also, I don't understand how an association can be highly significant, while simultaneously being at 'chance levels'.

The p-value is not saying as much as the log odds-ratio which is 1.16 vs 1.21, which are not entirely different. The p-value is testing if the log odds-ratio is different from 1.

Slight differences between the number of DMGs and the fact that there is an extreme response in gene expression in response to the loss of care within the FC population (i.e., many more DEGs) is ultimately making the FC comparison significant over the NC comparison. As we point out in the text, this amounts to an overlap of 49% (NC population) versus 51% (FC population) of DMGs and DEGs, which regardless of p-values seems like an irrelevant difference to us.

We have edited this line for clarity (now lines 204-208).

L221 The p-value in the evolved response is extremely low, yet you say that there is no association. Please provide a few words to explain what this test is doing.

We have edited this in the Results for clarity (now line 228).

The test is looking for a significant association/overlap between two categorical variables (DMGs vs DEGs). Significance can arise for high overlap/association ($\log(\text{OR}) > 1$) or a low/lack of association ($\log(\text{OR}) < 1$). We have clarified this in the methods section (now lines 575-577).

L 231 What is a 'fixed evolved difference'? A permanent change in methylation pattern? Do you really want to say 'fixed'?

Yes.

L352 Delete _

Done.

L361 Not sure what 'stable' means in this context. Across generations? Presence/absence is similar across individuals?

We have edited this sentence to clarify that we mean heritable (now lines 367-371).

L370 I think it would be clearer to simply say 'inheritance of the epigenetic change'.

Removed 'non-genetic'

L 399. Please provide references to this previous work.

We have added references to this line.

L 457. 'Each pair was placed in a separate breeding box.' Is repeated in the next sentence.

Deleted the first instance of this.

REVIEWERS' COMMENTS

Reviewer #1 (Remarks to the Author):

I think the authors responded to my comments well and in a comprehensive manner. I still think there are interesting questions to ask about uniquely methylated genes within a selection population, but it is not at the heart of the paper. It is more about the mechanistic association between methylation and gene expression. I think the paper will make a valuable contribution to the field regardless if the authors make further edits based on the responses below or not.

RtR comments.

L18 Response. No, it does not require orders of magnitude more samples in my view. Even if you wanted to be conservative, you could still ask how many methylated genes were observed in all the replicates of a single line not found in any other line. And, more importantly, to the major effect reported here do those genes also have reduced gene expression variance compared to its sister population of the same selection regime (or the other three populations). I do not think the authors get to have it both ways. 24 samples is presented as enough to estimate gene expression and methylation level of the selection history and current environment interaction, but then 12 samples is not enough to even characterize what is the pattern for uniquely methylated genes of each population. Even if it is not done to the same level of certainty, which it would not be because of the reduced samples, it is still a test of the certainty that increased relative methylation reduces gene expression variability.

L83 Response. I asked about relative gene expression not absolute gene expression. There is a universal association between methylation and absolute gene expression level, but there is no strong association between relative methylation level and relative gene expression across insect or arthropods (as stated). Relative gene expression is the context of the comment. That is also what you are measuring; if changes of methylation level within a locus are associated with changes of gene expression between two different selection lines and environments. Not just gains/losses are presented (Fig 3). I view Fig 2 as just important to understanding the association found here, and that is continuous data.

L101 Response. No, coming from different families of the same population (or selection lines) does not make them biological, nor technical replicates per se for that matter. It depends on the question being asked. Questions here are being asked about differences between selection regimes (and then responses to environments). The number of biological replicates is the number of lines.

I doubt the authors and I will agree on this. The other reviewer did not raise the issue. I think this paper is too interesting and adds to our understanding too much to be held up by the issue. Additionally, the

effect sizes are rather large and doing a nested analysis is unlikely to remove these effects, just make them less certain.

L409 Response. How are the parental beetles forced to care? Seems like they are just provided the opportunity to care? The authors cited work (another nice paper from this larger experiment) has ~6% of carcasses where both parents leave before larvae hatch – no parental care. So full care does not represent the same level of artificial selection as guaranteeing that parents will not get to interact with offspring. Especially because my understanding is that parents were not forced to interact.

Reviewer #2 (Remarks to the Author):

I am happy with the clarifications and revisions, and recommend that the paper be accepted for publication.

REVIEWERS' COMMENTS (Round 2)

Reviewer #1 (Remarks to the Author):

I think the authors responded to my comments well and in a comprehensive manner. I still think there are interesting questions to ask about uniquely methylated genes within a selection population, but it is not at the heart of the paper. It is more about the mechanistic association between methylation and gene expression. I think the paper will make a valuable contribution to the field regardless if the authors make further edits based on the responses below or not.

We thank the reviewer for their response and their assessment of our manuscript's significance. The remaining points raised are valuable, but, as argued below, we believe they are beyond the scope of the current manuscript and so do not require further edits.

RtR comments.

L18 Response. No, it does not require orders of magnitude more samples in my view. Even if you wanted to be conservative, you could still ask how many methylated genes were observed in all the replicates of a single line not found in any other line. And, more importantly, to the major effect reported here do those genes also have reduced gene expression variance compared to its sister population of the same selection regime (or the other three populations). I do not think the authors get to have it both ways. 24 samples is presented as enough to estimate gene expression and methylation level of the selection history and current environment interaction, but then 12 samples is not enough to even characterize what is the pattern for uniquely methylated genes of each population. Even if it is not done to the same level of certainty, which it would not be because of the reduced samples, it is still a test of the certainty that increased relative methylation reduces gene expression variability.

We apologise for any lack of clarity in our previous response. We accept the reviewer's point that it might be possible to carry out a rough analysis of stochastic changes in methylation simply comparing the two independent lines for each condition. However, it is simply not possible for us to do this with the data we generated. This is because the independent populations were pooled for methylation analysis (from a large number of individuals) so we only have multiple biological replicates per evolved condition (Full Care vs No Care). Thus, we can assay gene expression and variability within a condition with a reasonable amount of power (and, indeed, within an independent population) but not methylome variability between replicate populations. Moreover, we have pooled sufficiently across individual families/replicate lines to ensure these are representative of methylation status changes between populations (see Methods). Therefore, we focussed on population level changes for this particular analysis.

L83 Response. I asked about relative gene expression not absolute gene expression. There is a universal association between methylation and absolute gene expression level, but there is no strong association between relative methylation level and relative gene expression across insect or arthropods (as stated). Relative gene expression is the context of the comment. That is also what you are measuring; if changes of methylation level within a locus are associated with changes of gene expression between two different selection lines and environments. Not just gains/loses are presented (Fig 3). I view Fig 2 as just important to understanding the association found here, and that is continuous data.

We agree Fig 2 is just as important which is why we take the analysis approach we have in the paper. Given the association between expression and methylation and the mystery surrounding the role of DNA methylation in insects we think it is an obvious and important

question to ask about DNA methylation (both in terms of relative and absolute gene expression levels, as well as looking at gene expression variability) within the context of our experimental manipulations of care over several generations. For example, evolved changes in 5mC may be different to the dynamic changes observed in previous manipulations within an individual's lifetime (e.g., Cunningham et al., 2019 *J Exp Biol*).

L101 Response. No, coming from different families of the same population (or selection lines) does not make them biological, nor technical replicates per se for that matter. It depends on the question being asked. Questions here are being asked about differences between selection regimes (and then responses to environments). The number of biological replicates is the number of lines.

I doubt the authors and I will agree on this. The other reviewer did not raise the issue. I think this paper is too interesting and adds to our understanding too much to be held up by the issue. Additionally, the effect sizes are rather large and doing a nested analysis is unlikely to remove these effects, just make them less certain.

As the reviewer acknowledges, we do disagree on this point. The difference between biological and technical replicates can sometimes be semantic - however in this case what is more important is whether our approach is valid for the question that we ask. What we are trying to do is to estimate the variability in expression of each gene; this estimate of variability needs to capture genuine variability beyond noise introduced by sequencing technology and the use of read counts etc. Our approach, sampling multiple individuals from multiple families from each replicate population, will incorporate genuine variability in gene expression and methylation, because the different individuals differ both genetically and 'epigenetically' (i.e., these are biological replicates). The total variability in the expression will, as we did not have replicate measurements of exactly the same samples to compare, be a sum of the noise from the experimental process of measuring RNA levels and the genuine variability within and between populations. Moreover, siblings from the same families used in RNA sampling were pooled for methylation analysis, which further ensures that we are capturing the propensity for a gene to be methylated within and between populations (Full Care vs No Care). Thus, for the questions we are asking, our method clearly does give us estimates of variability in expression, using multiple biological replicates, which we can relate to the propensity of the gene to be methylated across populations.

L409 Response. How are the parental beetles forced to care? Seems like they are just provided the opportunity to care? The authors cited work (another nice paper from this larger experiment) has ~6% of carcasses where both parents leave before larvae hatch – no parental care. So full care does not represent the same level of artificial selection as guaranteeing that parents will not get to interact with offspring. Especially because my understanding is that parents were not forced to interact.

In the Full Care, there is no option to leave and perhaps we are not forcing them to care *per se*, but parents will interact more with their young than they might do naturally because they have no alternative in a confined breeding box (and offspring will solicit care through begging calls etc). We do not know precisely the nature of these interactions, but we cannot rule them out. Our original point was that in different ways, the Full Care and the No Care conditions created in the laboratory under artificial rearing conditions, will each differ from the natural condition in which the ancestral beetles evolved. Both will experience selection and drift to different degrees and for different reasons. Therefore, we have no compelling reason to agree with the reviewers' original assertion that Full Care solely experience drift.

Reviewer #2 (Remarks to the Author):

I am happy with the clarifications and revisions, and recommend that the paper be accepted for publication.